# Discovery and ramifications of incidental Magnéli phase generation and release from industrial coal-burning

Yi Yang [1,2,3], Bo Chen[4], James Hower[5], Michael Schindler [6], Christopher Winkler [7], Jessica Brandt [8], Richard Di Giulio[8], Jianping Ge[9], Min Liu[1], Yuhao Fu[10], Lijun Zhang[10], Yuru Chen[1], Shashank Priya [4] & Michael F. Hochella, Jr. [2,11]

Coal, as one of the most economic and abundant energy sources, remains the leading fuel for producing electricity worldwide. Yet, burning coal produces more global warming $CO_2$ relative to all other fossil fuels, and it is a major contributor to atmospheric particulate matter known to have a deleterious respiratory and cardiovascular impact in humans, especially in China and India. Here we have discovered that burning coal also produces large quantities of otherwise rare Magnéli phases ($Ti_xO_{2x-1}$ with $4 \leq x \leq 9$) from $TiO_2$ minerals naturally present in coal. This provides a new tracer for tracking solid-state emissions worldwide from industrial coal-burning. In its first toxicity testing, we have also shown that nanoscale Magnéli phases have potential toxicity pathways that are not photoactive like $TiO_2$ phases, but instead seem to be biologically active without photostimulation. In the future, these phases should be thoroughly tested for their toxicity in the human lung.

[1] Key Laboratory of Geographic Information Science of the Ministry of Education, School of Geographic Sciences, East China Normal University, 500 Dongchuan Road, Shanghai 200241, China. [2] Department of Geosciences, Virginia Tech, Blacksburg, VA 24061, USA. [3] State Key Laboratory of Estuarine and Coastal Research, East China Normal University, 3663 North Zhongshan Road, Shanghai 200062, China. [4] Center for Energy Harvesting Materials and Systems, 310 Durham Hall, Virginia Tech, Blacksburg, VA 24061, USA. [5] Center for Applied Energy Research, 2540 Research Park Drive, Lexington, KY 40511, USA. [6] Department of Earth Sciences, Laurentian University, Sudbury, ON, Canada P3E 2C6. [7] Nanoscale Characterization and Fabrication Laboratory, Institute for Critical Technology and Applied Science, Virginia Tech, Blacksburg, VA 24061, USA. [8] Nicholas School of the Environment, Levine Science Research Center, Duke University, Durham, NC 27708-0328, USA. [9] Shanghai Key Laboratory of Green Chemistry and Chemical Processes, Department of Chemistry, East China Normal University, 3663 North Zhongshan Road, Shanghai 200062, China. [10] Key Laboratory of Mobile Materials MOE, State Key Laboratory of Superhard Materials, and Department of Materials Science, Jilin University, Changchun 130012, China. [11] Geosciences Group, Energy and Environment Directorate, Pacific Northwest National Laboratory, Richland, WA 99352, USA. Yi Yang and Bo Chen contributed equally to this work. Correspondence and requests for materials should be addressed to M.F.H. (email: hochella@vt.edu)

Coal has been used by humans for thousands of years to produce energy through its combustion. According to the International Energy Agency[1], roughly 30% of the world's overall energy needs come from coal. The energy/industrial impact of coal is probably best illustrated in this way: it is used to produce approximately 40% of the world's electricity, more than any other energy source, and 70% of the world's steel. Further, its use for these purposes has continually increased over the last 150 years. At current usage rates, and considering global reserves and demand in the developing counties, especially India, coal will likely be a significant player in the world's energy portfolio well into the future.

Nevertheless, dealing with the downsides to burning coal on massive, protracted scales are formidable, including both severe environmental and human health impacts. While greenhouse gas drivers of climate change[2] are the most important consequence to address long-term, the most important short-term consequence is that coal-burning is a major contributor to atmospheric particulate matter with aerodynamic diameter smaller than 2.5 μm (so-called PM2.5), which is known to have a significant impact on human health[3–5]. Air pollution, of which PM2.5 is the most important fraction vs. larger fractions, has been estimated to lead to 3.3 million premature deaths per year worldwide. In China alone, 1.6 million premature deaths are estimated annually due to cardiovascular and respiratory injury over time[4,5]. Most Chinese megacities suffer over 100 severely hazy days every year with PM2.5 concentrations two to four times higher than World Health Organization (WHO) guidelines[6]. Coal combustion is one of the major contributors to PM2.5 in China. Attention has also been given specifically to ultrafine PM because these smallest (nanoscale) particles can penetrate the lung's alveolar (air-sack) membranes and thereafter translocate into the bloodstream to other organs, including the heart, and many are also capable of passing the blood-brain barrier[7].

Over the last 50 years, as world coal use continues to increase as one of the most economic, abundant, and utilized fuel sources, in parallel, environmental scientists have also been discovering the severe detrimental aspects of this particular energy source. At the same time, a new and dramatically important revolution in science and technology has also been developing, that of nanoscience and nanotechnology, which expedited these environmental discoveries. Especially in the last 20 years, the study of naturally-occurring and "incidental" nanomaterials has become an important subset of the nanoscience and technology revolution[8,9]. In particular, incidental nanomaterials are produced unintentionally (more specifically, an unintended nano-byproduct of some anthropogenic process), and as a result commonly enter the "environmental world" very soon or immediately upon formation. Examples include direct producers of incidental nanomaterials (e.g., engines and sewage treatment plants) and indirect producers of incidental nanomaterials (desertification and mining, where humans disturb the Earth's surface, and natural processes respond in part by making incidental nanomaterials, i.e., nanomaterials that nature would not have produced otherwise). Many of these natural and incidental nanomaterials possess unique properties, many not shown by their bulk equivalents, and they can contribute disproportionately to environmental chemistry[10]. Until relatively recently, incidental nanomaterials went unnoticed or were without concern. Now, with the development of environmental nanoscience and technology, this is more often no longer the case.

It is the rise of coal, and the rise of incidental environmental nanoscience, that have come together to create the study presented in this article, that is, the discovery of titania suboxides, specifically Magnéli phases ($Ti_xO_{2x-1}$), as an incidental nanomaterial consequence of coal combustion. We determined in this study that $Ti_xO_{2x-1}$ is produced in the coal combustion process from stoichiometric titanium oxide, $TiO_2$, which very commonly occurs as an accessory (minor, by abundance) mineral in many coals with different coal ranks and sources worldwide[11–16].

We originally made the discovery of incidental Magnéli phases during an investigation of a coal ash spill into a riverine environment[17]. In that study, we presumed its generation was associated with coal combustion (more detail is given about this discovery in the Results and Discussion sections below). In the process of following the associations and mobility/bioavailability of arsenic due to this coal ash spill, we observed and recognized these Magnéli phases, which until now were only very rarely observed in nature, but on the other hand are much more commonly synthesized and studied by materials scientists because of their unusual properties. We did not mention our discovery of these titania suboxides in Yang et al.[17], because so much more verification and assessments of them needed to be completed at that time.

In this article, we describe in detail how we recognized and categorized these incidental Magnéli phases, our exact crystallographic analyses of the various suboxide varieties encountered, our syntheses of these phases which strongly supports our hypothesis that these materials are generated during coal combustion in power plants, why Magnéli phases are highly reliable tracers for determining coal ash distributions via aeolian or riverine processes in the environment, and finally a first look at its toxicity using the well-established zebrafish (*Danio rerio*) embryo model.

## Results

**Initial discovery of environmental Magnéli particles**. Sediment samples were collected upstream and downstream from the Dan River (North Carolina, USA) coal ash spill site[17] (also, see the introduction to this paper above), on 9 February 2014. In addition, coal ash samples still in the plant impoundment pond, as well as unburned coal still at the plant site, were collected. Based on $HNO_3$–$HClO_4$–HF digestion followed by inductively coupled plasma mass spectrometer (ICP-MS) analyses of these samples, titanium concentrations are 0.28–0.42 g kg$^{-1}$ (in the unburned coal sample), 6.1–6.6 g kg$^{-1}$ (in the coal ash samples from the plant impoundment), 4.6–5.2 g kg$^{-1}$ (in river sediments upstream from the spill site), and 4.7–6.1 g kg$^{-1}$ (in river sediments downstream from the spill site). Compared to Ti concentrations of unburned coal vs. coal ash, and slightly elevated Ti concentrations in the downstream sediment samples vs. upstream, the data suggest that coal ash is a source for elevated Ti in the downstream sediments.

In addition to the $TiO_2$ polymorphs, anatase and rutile (Supplementary Fig. 1), a different Ti oxide phase ($Ti_xO_{2x-1}$) was noticeable in the downstream sediment samples, but never observed in the upstream sediment samples. These downstream $Ti_xO_{2x-1}$ particles are in the size range of a few tens of nm to hundreds of nm, showing *d*-spacings in electron diffraction patterns of neither anatase nor rutile. Based on the wavy electron density contrast and ultrafine linear striations in bright field imaging (Fig. 1), the Magnéli phases of Ti oxides, which conform to the $Ti_xO_{2x-1}$ formulation, were suggested. Anatase and rutile never exhibit the characteristic linear striations of the titania suboxides. These titania suboxides also have *d*-spacings larger than any possible from anatase or rutile (see next two sections below for specific phase identification).

**Magnéli phases and their specific identification**. What would eventually become known as Magnéli phases were first synthesized in the 1930's by materials scientists[18]. These unique phases

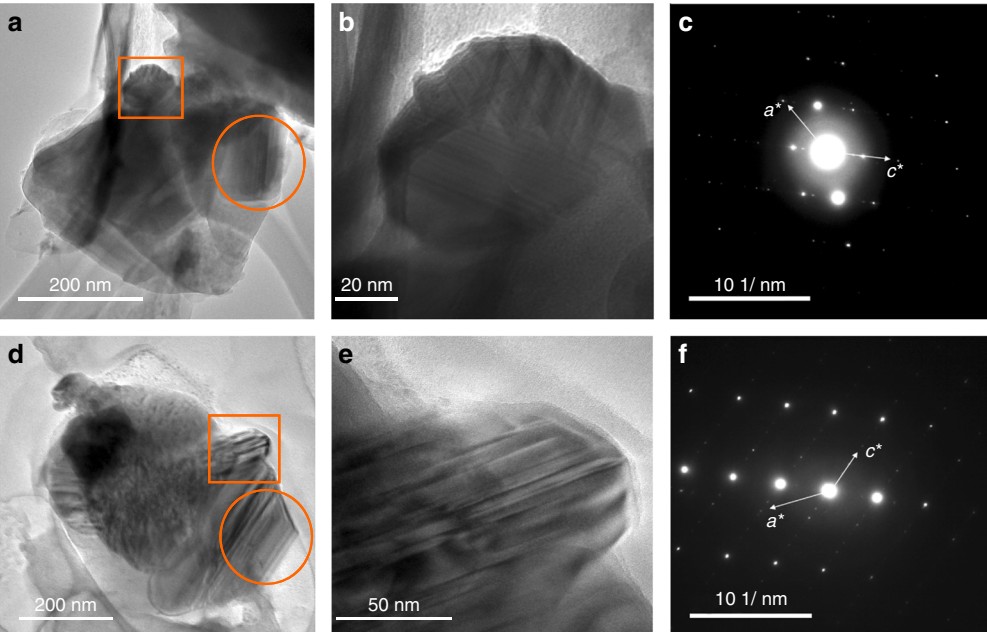

**Fig. 1** Identification of Magnéli phases in environmental and industrial samples. **a** TEM image of $Ti_xO_{2x-1}$ in a Dan River downstream sediment sample. **b** A magnified TEM image of the square selected area in **a**. **c** Selected area electron diffraction (*SAED*) pattern of the circular selected area in **a**. Vectors $a^*$ and $c^*$ denote two of the principal crystallographic axes of Magnéli phases in reciprocal (diffraction) space (see Fig. 2 and associated text). **d** TEM image of $Ti_xO_{2x-1}$ in coal ash collected from the coal ash impoundment. **e** A magnified TEM image of the square selected area in **e**. **f** SAED pattern of the circular selected area in **d**

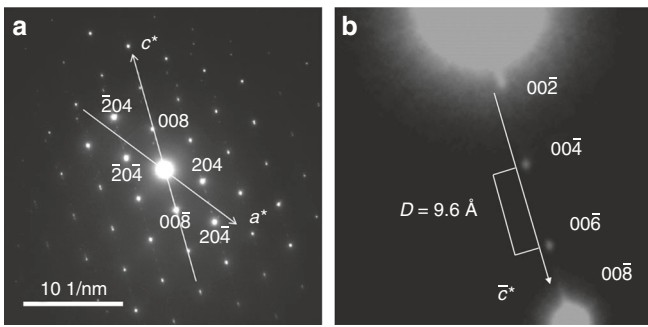

**Fig. 2** Identification of specific Magnéli phases using electron diffraction. **a** SAED pattern from $Ti_6O_{11}$. **b** A magnified image of diffraction spots along $c^*$ displaying a repeat distance of 9.6 Å, which is characteristic of the Magnéli phase $Ti_6O_{11}$. See text and Supplementary Table 1 for details

have been produced experimentally by reducing $TiO_2$ at low oxygen fugacities[19, 20] (e.g., using a hydrogen atmosphere) and at low confining pressures[21] (1 bar or 1 atm) over a broad range of high temperatures (900–1700 °C). Wadsley[22] and Andersson et al.[23]. first recognized these titania suboxides as derivatives of the rutile ($TiO_2$) structure with certain oxygen atoms removed along the crystallographic shear plane (121). Variations in the Magnéli family have the formula $Ti_xO_{2x-1}$ with $4 \leq x \leq 9$. To be complete, although not seen in this particular study, there is another family of titania suboxides with the same general formula as the Magnéli phases except with $16 \leq x \leq 36$ (e.g., see ref. [24]). This newer class of titania suboxides have oxygen atoms removed along the crystallographic shear plane (132). In both types of these triclinic shear structures, $TiO_6$ octahedra that are normally linked by corner and edge sharing in the structure of rutile, instead share edges and faces along one of the two crystallographic shear planes[25, 26].

Conveniently, the structures between different members of the Magnéli series can be directly compared with each other and with

the structure of rutile if one uses the crystallographic model proposed by Le Page and Strobel[27]. In this model, the *c*-axis in the Magnéli phase is parallel to the *c*-axis in the rutile structure (i.e., parallel to a chain of edge-shared $TiO_6$ octahedra), and the *b*-axis in the Magnéli phase is parallel to the crystallographic shear plane axis [121]. This setting allows distinguishing between different members of the homologous series $Ti_xO_{2x-1}$ within $4 \leq x \leq 9$ on the basis of the magnitude of the *c*-axis, as the *a*- and *b*-axes are roughly similar in length across the family (Supplementary Table 1). In selected electron diffraction patterns taken from crystal orientations parallel to the ac-plane (101) or bc-plane (011), the orientation of the reciprocal $c^*$ axis can be easily identified via the small $d^*$-spacing between diffraction spots. Measurement of *d*-spacings along this axis allows for a rapid identification of an individual member of the Magnéli family (Fig. 2a, b).

**The widespread presence of incidental Magnéli phases**. With Magnéli phases discovered in coal-derived ash at the Dan River power station site, as well as in downstream river sediment impacted by the ash spill there, the next step in this study was to see if this family of titania suboxides was inherent to this particular plant, whether these phases are a ubiquitous feature of coal-burning power plants in general, or whether the reality is somewhere in between these two end-member scenarios. Although an exhaustive study will take much longer, we selected 21 additional ashes for testing. The coal ash samples were taken from 12 coal-burning power plants utilizing various coals mined in a number of areas in the United States and China. Specific information on the coal ash analyzed with transmission electron microscopy (TEM) is summarized in Table 1.

One to three Magnéli phases, within the known compositional range between $Ti_4O_7$ and $Ti_9O_{17}$, were found in all 21 additional samples investigated using TEM. Three examples of these Magnéli phases are shown in Fig. 3; two of these samples are

**Table 1 Coal ash samples tested for occurrence of Magnéli phases using TEM**

| No. | Coal ash sample | Coal origin/information | Observed Magnéli phase(s) |
|---|---|---|---|
| 1 | Dan River coal ash | Coal ash impoundment at Dan River, North Carolina, USA; coal origin: unknown | $Ti_6O_{11}$, $Ti_7O_{13}$ |
| 2 | Virginia bottom ash | Coal power plant, southwest Virginia, USA; coal origin: unknown | $Ti_6O_{11}$, $Ti_8O_{15}$, $Ti_5O_9$ |
| 3 | Virginia fly ash | Coal power plant, southwest Virginia, USA; coal origin: unknown | $Ti_6O_{11}$, $Ti_8O_{15}$, $Ti_5O_9$ |
| 5 | Kentucky plant M 2007 fly ash | Illinois Basin, USA | $Ti_7O_{13}$ |
| 6 | Kentucky plant H 2007 fly ash | Central eastern Kentucky, USA; low-sulfur coal | Magnéli phase(s) observed but not specifically identified |
| 7 | Kentucky plant H 2015 fly ash | Western Kentucky, USA; high-sulfur coal | Magnéli phase(s) observed but not specifically identified |
| 8 | Kentucky plant I 2001 fly ash | Southeastern Kentucky, USA; low-sulfur coal | $Ti_9O_{17}$, $Ti_8O_{15}$ |
| 9 | Kentucky plant I 2001 bottom ash | Southeastern Kentucky, USA; low-sulfur coal | $Ti_6O_{11}$ |
| 10 | Kentucky plant I 2007 fly ash | Southeastern Kentucky, USA; low-sulfur coal | $Ti_6O_{11}$ |
| 11 | Texas power plant fly ash | Texas Gulf Coast, USA; Eocene coal | $Ti_6O_{11}$/$Ti_7O_{13}$ |
| 12 | Texas power plant bottom ash | Texas Gulf Coast, USA; Eocene coal | $Ti_6O_{11}$/$Ti_7O_{13}$ |
| 13 | New Mexico power plant fly ash | New Mexico San Juan Basin, USA; Cretaceous coal | $Ti_6O_{11}$/$Ti_7O_{13}$ |
| 14 | New Mexico power plant bottom ash | New Mexico San Juan Basin, USA; Cretaceous coal | $Ti_4O_7$ |
| 15 | Missouri power plant fly ash | Wyoming Powder River Basin, USA; Paleocene coal | Magnéli phase(s) observed but not specifically identified |
| 16 | Missouri power plant bottom ash | Wyoming Powder River Basin, USA; Paleocene coal | Magnéli phase(s) observed but not specifically identified |
| 17 | China SZDC 2-1 | Chongqin, China; anthracite, Permian coal | $Ti_9O_{17}$ |
| 18 | China SZDC 2-2 | Chongqin, China; anthracite, Permian coal | $Ti_6O_{11}$, $Ti_7O_{13}$ |
| 19 | China DD 2-1 | Yunnan, China; anthracite | $Ti_6O_{11}$ |
| 20 | China DD 2-2 | Yunnan, China; anthracite | — |
| 21 | Xuzhou, China fly ash | Coal origin: unknown | $Ti_6O_{11}$, $Ti_8O_{15}$, $Ti_5O_9$ |
| 22 | Shanghai, China fly ash | Coal origin: unknown | $Ti_5O_9$, $Ti_6O_{11}$ |

from the United States, and one is from China. Additional examples are shown in Supplementary Fig. 2. All Magnéli phases found are characterized by their distinctive superfine striation patterns (shown in both light and dark field imaging in Fig. 3) and their corresponding diffraction patterns which show long repeats (in real space) along $c^*$.

Overall, the most common phase observed in ash samples collected from all 22 locations is $Ti_6O_{11}$ (see also Table 1), suggesting that this phase is predominantly emitted by the coal-burning power plants relative to other members of the $Ti_xO_{2x-1}$ ($4 \leq x \leq 9$) family.

If titania suboxides are so commonly present in coal ash, including fly ash, we began looking for these phases in a much broader sense in the environment by considering aeolian transport processes in power plant stack emissions (these processes are discussed in the Discussion section below). Supplementary Table 2 lists five locations where we searched for these titania suboxides with TEM, including sediments from a lake that at one time received coal ash pond effluent, and from an urban stormwater pond that never directly received coal ash pond effluent, both in USA locations, and in a Chinese estuary not directly impacted by coal ash spills (to our knowledge), sludge from a waste water treatment plant in China, as well as road dust in a Chinese urban area. In each case, we were successful in finding titania suboxide grains that had the characteristic linear nano-fine striations of these suboxides. Several examples are shown in Supplementary Fig. 3.

**Synthesis of Magnéli phases under coal combustion conditions.** Due to the fact that titania suboxides have never been confirmed to occur in coal, have never been reported in the ecological environment, and have only been very rarely observed in geological settings (specifically in rocks, having either an extraterrestrial/meteoritic origin, or a very unusual

high temperature—roughly 1200 °C—and highly reducing rock-forming scenario; see Discussion section below), it seemed likely that Magnéli phases were being generated in coal-burning power plants. In this regard, we hypothesized that it was most likely that naturally occurring $TiO_2$ coal-accessory minerals (the polymorphs rutile, anatase, brookite) were being converted to Magnéli phases in the high temperature coal combustion chambers of coal fired power plants. To test this hypothesis, we conducted an extensive set of experiments and measurements as described here.

Magnéli phases were generated in our laboratories by heating pulverized coal with anatase, rutile, or commercial "P25" nanoparticles (a mixture of approximately 80% anatase, 20% rutile) under an $N_2$ atmosphere for two hours (Fig. 4). In these first experiments, our experimental set-up was not meant to closely simulate combustion chambers in actual power plants which would be difficult, but to give a similar, high temperature, low available oxygen effect. The combustion chambers of coal-burning power plants are operated with pulverized coal in the presence of ambient air designed to produce very high temperatures (1400–1500 °C) with the coal carbon and the $O_2$-airborne gas converted to $CO_2$, leaving very hot $TiO_2$ temporarily in an oxygen poor environment. Our set-up allowed us to externally heat the titania at well controlled temperatures up to 1100 °C in the presence of $N_2$ gas, oxidizing a very small amount of the accompanying powdered coal (to $CO_2$) using only oxygen derived from traces of $O_2$ in the nitrogen environment, plus the de-oxygenating titania. Under these conditions, after annealing P25 $TiO_2$ at 550 °C, the color of the P25 powder changed from white to gray, and the color of coal-annealed P25 $TiO_2$ turned to black when the annealing temperature was higher than 700 °C (Fig. 4b). The X-ray diffraction (XRD) pattern of coal-annealed P25 did not change up to an annealing temperature of 800 °C; interestingly, new characteristic XRD

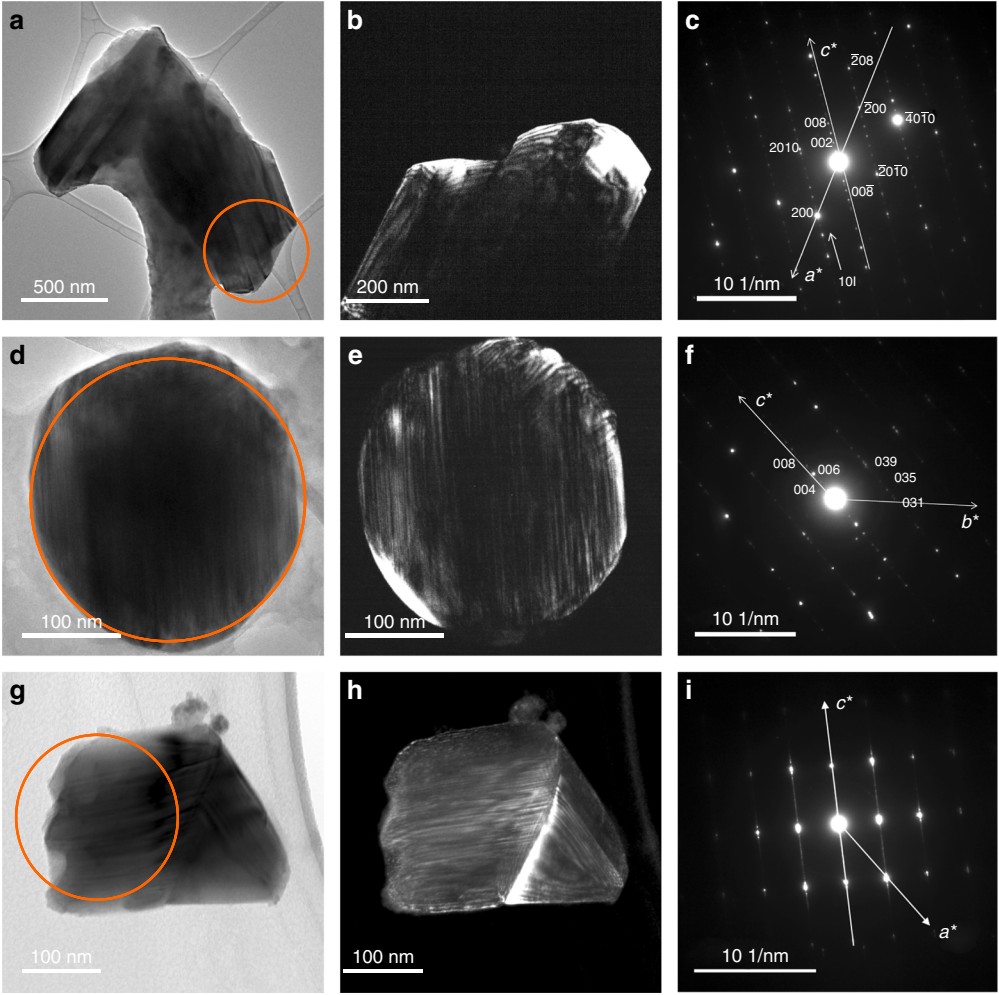

**Fig. 3** Three examples of Magnéli phases of titania suboxides identified in coal ash from US and Chinese coals. **a–c** A $Ti_xO_{2x-1}$ particle in bottom ash taken from a coal-fired power plant I, Kentucky, USA, in 2001 (No. 9 in Table 1): **a** bright field TEM image, **b** dark field TEM image, and **c** SAED pattern of the selected area in **a**, indicating a $Ti_6O_{11}$ particle. **d–f** A $Ti_xO_{2x-1}$ particle in fly ash taken from a coal-fired power plant in Xuzhou, China, in 2015 (No. 21 in Table 1): **d** bright field TEM image, **e** dark field TEM image, and **f** SAED pattern of the selected area in **d**, indicating a $Ti_6O_{11}$ particle. **g–i** A $Ti_xO_{2x-1}$ particle from fly ash taken from a southwest Virginia, USA, coal-fired power plant, USA, in 2014 (No. 3 in Table 1): **g** bright field TEM image, **h** dark field TEM image, and **i** SAED pattern of the selected area in **g**, indicating a $Ti_8O_{15}$ particle

peaks emerged when the annealing temperature was at 900 °C or above (Fig. 4c). Compared with diffraction peaks reported in International Centre for Diffraction Data (ICDD) files, as well as TEM observations (Supplementary Fig. 4), we found that the dominant phase for converted P25, heated in the presence of coal powder at 900 °C (designed P25-900 °C), is the $Ti_6O_{11}$ Magnéli phase, and the dominant phase for coal-annealed P25 at 1100 °C (designed P25-1100 °C), is the $Ti_4O_7$ Magnéli phase. Supplementary Table 3 shows that other Magnéli phases are also produced at these temperatures and conditions, and that grains of rutile and anatase can remain unconverted, at least after the 2-h duration of these experiments. Moreover, the Raman spectra in Supplementary Fig. 5a also show that the coal-annealed P25-900 °C and P25-1100 °C samples produced new and distinctive peaks compared with pristine P25 $TiO_2$ powder, although the generation of Magnéli phases when going from 700 to 900 °C is not as apparent in the Raman spectra. When we switched the starting titania material from P25 $TiO_2$ to anatase or rutile $TiO_2$, the same distinguishing XRD peaks for various Magnéli phases were observed when annealing temperatures were the same as previously defined (Supplementary Fig. 5b, c). This demonstrates that the 900 °C temperature region is the

approximate minimum temperature, which can be used to generate Magnéli phases by combusting $TiO_2$ with coal under a nitrogen environment.

**Characterization of synthesized Magnéli phases**. The thermal stability of Magnéli phases in air is a key characteristic of these materials very much relevant to this study. We explored this property using thermogravimetric analysis (TGA) and XRD. As shown in Supplementary Fig. 6a, b, black Magnéli P25-1100 °C samples do not show any obvious phase change or weight change during heating at 150 and 250 °C for 12 h. Further, we found that the black Magnéli P25-1100 °C sample demonstrated excellent thermal and XRD pattern stability at 150 and 250 °C over times of 7 and 15 days (Supplementary Fig. 6c, d). However, when the temperature is increased to 350 °C, the black Magnéli powder changes color to a gray powder, and rutile peaks appear in the XRD patterns; at 450 °C for 12 h, the black Magnéli powder became a yellowish color and was completely converted to rutile, accompanied by a weight increase of 3.85%.

The ultraviolet (UV)–visible (Vis) absorption spectra in Supplementary Fig. 7a indicates that the black P25-900 °C and

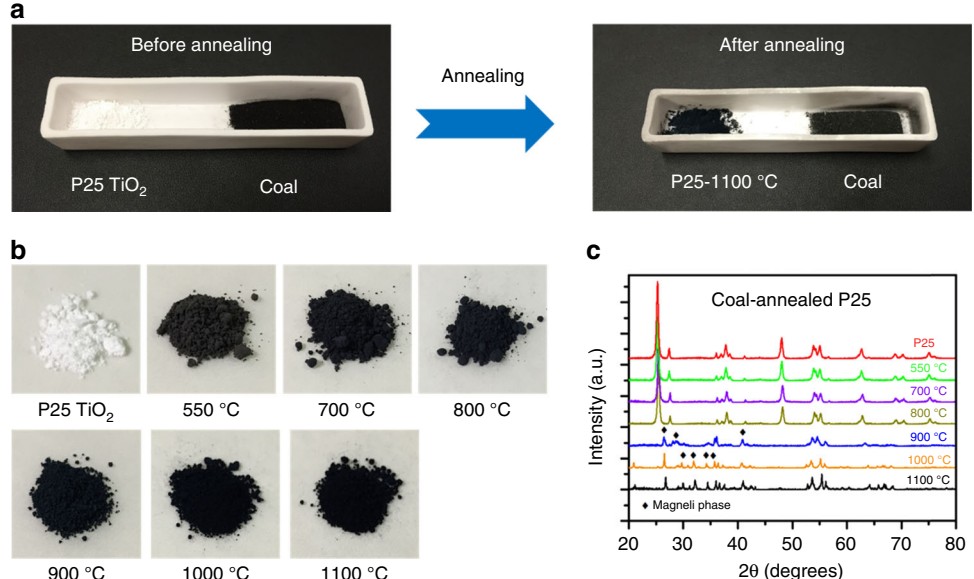

**Fig. 4** Laboratory synthesis of Magnéli phases from $TiO_2$ under high temperature, low-oxygen conditions. **a** Synthesis of Magnéli $Ti_xO_{2x-1}$ by annealing P25 $TiO_2$ nanoparticle with coal in a pure $N_2$ atmosphere. See text for details. **b** Photograph of different titanium oxides after annealing P25 $TiO_2$ nanoparticle with coal at different temperatures. **c** XRD patterns of coal-annealed P25 $TiO_2$ at different temperatures

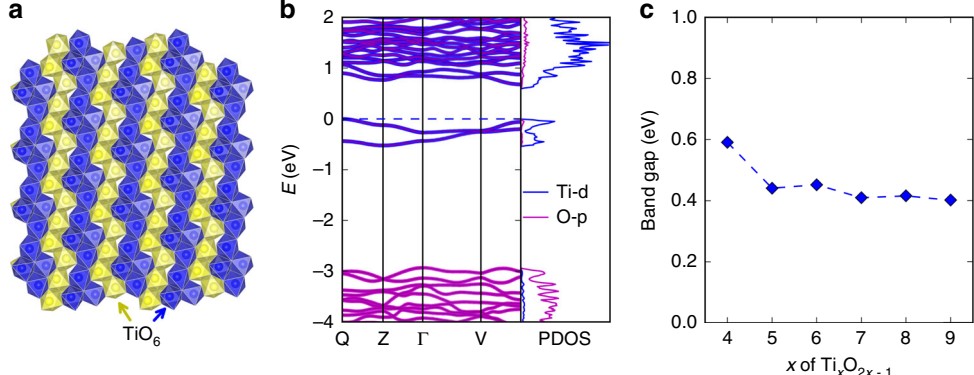

**Fig. 5** Structural and calculated electronic properties of Magnéli phases. **a** Schematic illustration of the crystalline structure of $Ti_4O_7$. See text and Supplementary Fig. 8. **b** Calculated band structure and (orbital) projected density of states (*PDOS*) of $Ti_4O_7$. **c** Calculated band gaps of the Magnéli phases $Ti_xO_{2x-1}$ as the function of $x$

P25-1100 °C samples have excellent light absorption in the UV, visible, and near-infrared light range. The electrical resistance of P25-900 °C and P25-1100 °C is about 80 and 7 Ω cm, respectively, at room temperature, which is roughly $10^5$ times lower than pristine P25.

Even though the black Magnéli phase $Ti_xO_{2x-1}$ powders have good light absorption properties as demonstrated in Supplementary Fig. 7a, the Magnéli phase powders do not display any photocatalytic performance as indicated by their inability to degrade methylene blue under simulated sunlight conditions (Supplementary Fig. 7b, c). Coal-annealed anatase-900 °C does show some amount of photocatalytic activity, but this is likely due to remnant anatase in the sample, as can be seen in the XRD pattern in Supplementary Fig. 5b. When only visible light was used as the light source, the coal-annealed anatase-900 °C sample and all other Magnéli phases did not demonstrate any obvious photocatalytic performance.

To gain insights into the optoelectronic properties of the Magnéli phases, we performed first-principle band structure calculations using the density functional theory (DFT) method. The crystalline structures of $Ti_4O_7$ and other Magnéli phases,

which are composed of chains of edge-sharing $TiO_6$ octahedra interrupted every $x^{th}$ octahedron, are illustrated in Fig. 5a and Supplementary Fig. 8. The calculated band gap of $Ti_4O_7$ is 0.59 eV (Fig. 5c), and the gap values of $Ti_xO_{2x-1}$ show a decrease with $x$. The band gaps are opened within Ti-$d$ states (as demonstrated by the orbital-projected density of states in Fig. 5b), caused by the formation of the dimerization pattern of $Ti^{3+}$-$Ti^{3+}$ pairs[28]. The generally small band gaps of $Ti_xO_{2x-1}$ from calculations are consistent with the excellent light absorption characteristic of these materials.

**Initial toxicity study of Magnéli phases**. The observations and experiments described above show that $Ti_6O_{11}$ is the most commonly observed Magnéli phase in our suite of coal ash samples, and that $Ti_6O_{11}$ is also the dominant phase in our 900 °C synthesis experiments. Therefore, the toxicity experiment reported in this paper was carried out using the P25 samples prepared at 900 °C (P25-900 °C), and compared with untreated P25, as well as samples with no titanium oxides added as control treatment groups.

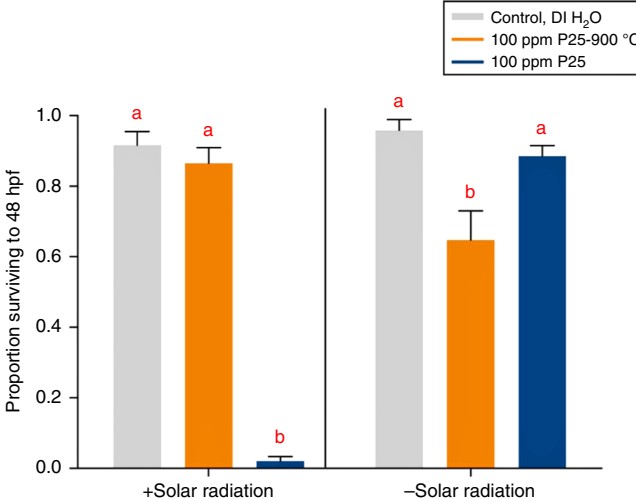

**Fig. 6** P25 or P25-900 °C nanoparticle toxicity when exposed to dechorionated zebrafish embryos. All measurements are made 48 h post fertilization (hpf), under co-exposure to simulated solar radiation, SSR (labeled " + Solar radiation") or without SSR co-exposure (labeled "-Solar radiation"). Titania (P25 or P25-900 °C) exposures were made at 100 ppm, and compared to control groups at 0 ppm titania. Letters "a" and "b" signify statistically different responses among nanomaterial treatments within the + SSR and − SSR co-exposure groups ($P < 0.05$), with error bars depicted here as plus or minus the standard error of the mean

Figure 6 shows dechorionated zebrafish embryo survival at 48 h post fertilization (hpf) in replicates exposed to P25 nanoparticles with and without concurrent simulated solar radiation (SSR), at titania concentrations of 100 ppm, as well as control groups with no titania exposure, all this modeled after Bar-Ilan et al.[29] for the sake of comparison. The only difference between the Bar-Ilan et al. study and the present one is that exposure to P25-900 °C nanoparticles was also carried out.

Embryos were dechorionated at 24 hpf and, in the presence of SSR, immediately exposed to 100 ppm P25 or P25-900 °C, or no nanoparticles (the control experiments), with survival assessed at 48 hpf. The embryos co-exposed to P25-900 °C + SSR were not different from controls, but those exposed to P25 + SSR exhibited very high mortality (adjusted $P$-value < 0.0001). In the absence of SSR, we observed a very different result. There, dechorionated embryos exposed to 100 ppm P25-900 °C showed significant mortality (adjusted $P$-value = 0.0004), but those exposed to P25 showed slightly reduced, but not significantly different survival (adjusted $P$-value = 0.8626). These results (absence of SSR) exhibit the opposite trends as seen in the SSR-exposed experiments.

### Discussion

Our hypothesis is that the great majority of Magnéli phases in the environment are derived from $TiO_2$ minerals present in coal. This is strongly suggested by both the circumstantial evidence presented above concerning the spatial and temporal relationships between Magnéli phases and coal-fired power plants (over 1000 °C, locally reduced environments resulting from areas of incomplete combustion in the firebox), as well as the synthesis of Magnéli phases under the conditions of our heating experiments (over 900 °C, $TiO_2$ with pulverized coal under a $N_2$ atmosphere). It is also important to consider the possibility of generating Magnéli phases during coke production. Coke, typically made under highly reducing conditions from low-sulfur, low-ash bituminous coal at temperatures similar to those used in this study (over 900 °C), should result in Magnéli phases production.

As mentioned in the Introduction, coke made from coal is used to produce 70% of the world's steel.

$TiO_2$ naturally occurs in coal as anatase, rutile, and brookite, the three mineral $TiO_2$ polymorphs, whereas Magnéli phases have never been reported in any rank of coal, or coal origin. In summary, Swaine[30] summarized Ti associations in coals, noting that Eskanazi[31] called for an organic association of Ti. While an organic association may be the case in peats and low-rank coals, Finkelman[11] found anatase and rutile in bituminous coals. Ketris and Yudovich[12] acknowledge this mixed affiliation, placing Ti among the weakly or moderate coalphile elements. The levels of $TiO_2$ in coal vary, but generally are found within a range from roughly a few tenths of a percent, to a few percent, by weight. For example, Hower and Bland[13] and Hower et al.[14] found the $TiO_2$ content in the Pond Creek coal in eastern Kentucky to vary between 0.13–6.52%, with the highest amounts found in association with relatively high concentrations of Zr (as zircon, $ZrSiO_4$) in clastic-origin lithotypes. Similar trends were noted in eastern Kentucky's River Gem[15] and Elswick[16] coals.

It should be noted that we have considered the possibility that Magnéli phases result from $TiO_2$ polymorphs, especially anatase, due to its common use in the catalytic beds of modern coal-fired power plants in the combustion gas exhaust stream. These beds are designed to minimize/eliminate $NO_x$ in the exhaust stream, commonly using anatase/rutile to provide favorable chemical conversion of these toxic emission gases present depending on the type of coal that is burned. Any release of the titanium oxides in the gas stream during years of service would be trapped by the plant's electrostatic precipitators along with the fly ash exhaust component that it is designed to collect. However, such a scenario does not account for two factors. First, anatase/rutile used in these catalytic beds (e.g., P25) lack Magnéli phases. Second, the temperature of gases in this part of the power plant are many hundreds of degrees too low to generate Magnéli phases. Temperatures of at least 900 °C were needed in this study to convert anatase/rutile to Magnéli phases, with that temperature easily exceeded in the combustion chamber of a power plant. Therefore, $TiO_2$ used in catalytic beds should not be a source for titania suboxides from coal-burning power plants.

It should also be noted that burning municipal solid waste is becoming more and more common worldwide, and the production and/or fate of nanoparticles (including $TiO_2$ in commercial products) during this incineration is of growing interest (e.g., refs [32, 33]). Many incineration protocols use temperatures below what we have found to be conducive to Magnéli phase production under highly reducing conditions (<900 °C). In addition, our preliminary experiments to date using P25 starting material in the presence of sewage sludge under $N_2$ at 900 and 1000 °C produced rutile exclusively (as measured by powder XRD). However, some incineration is driven by burning coal at temperatures well in excess of 1000 °C, and here we would expect the production of Magnéli phases both from the $TiO_2$ in the coal itself and any in the waste. Given a 2014 estimate that 8600 metric tons of nanomaterials are incinerated annually on a global basis[34], and that $TiO_2$ nanomaterials would be some fraction of this, and that high temperature incineration using a carbon-based fuel would be some fraction of this, the production of Magnéli phases due to incineration is several orders of magnitude less than our estimated Gt levels of Magnéli phase generation due to industrial coal-burning (see Discussion below). Therefore, waste incineration is not expected to be a significant contributor to Magnéli phase production relative to coal-burning power plants.

Schindler and Hochella[35] very recently identified Magnéli phases in soils proximal to Cu-Ni-sulfides smelters and refineries in Sudbury, Ontario, Canada. Although the identified nano-size Magnéli phases only occurred as minute traces of material in

mineral surface coatings, its discovery in this setting suggests that these phases can also form during the reductive smelting and refining of metal-bearing ores in the presence of coke, coal, or carbon monoxide. In this scenario, this also opens the possibility of Magnéli phases originating from $TiO_2$ polymorphs present in the ore being smelted and refined. Hence, the anthropogenic formation of Magnéli phases may not be limited to coal-burning power plants and likely also coking facilities. Nevertheless, further study is needed, and this paper has concentrated on coal-burning power plants due to their tremendous and still growing abundance in the industrial world.

With the seeming ubiquity of incidental Magnéli phases from coal combustion, in fact it is quite the opposite for naturally occurring Magnéli phases. Such naturally occurring instances are very rarely recognized. Magnéli phases have infrequently been observed in lunar rocks, as oxygen deficient, blue "rutile"[36, 37]. Further, Magnéli phases have been suspected or identified, for example, as rare Ti oxide phases in interplanetary dust particles[38], in carbonaceous chondrite meteorite matrices[39], and in micro-meteorites in Antarctic ice particles[40]. Aside from these extra-terrestrial origins, the only Earth-bound natural occurrence that we could locate in the literature comes from Pedersen and Rønsbo[41] who discovered "mudstone xenoliths" exposed in one location on the central coast of western Greenland where andesitic magma likely flowed into shallow water underlain by a carbonaceous mudstone, resulting in high temperature (about 1200 °C), low confining pressure (close to five bars), extremely reducing (producing native iron), pyrometamorphosed sediment xenoliths where anatase/rutile grains were transformed to Magnéli phases.

Given our assessment that the overwhelming majority of nanoscale Magnéli phases found in the environment are from industrial coal-burning, it is possible that these Magnéli phases could be used as a tracer for the global distribution of coal-burning solid-state emissions. Scientists have been working on using various source apportionment techniques for tracking coal-burning emissions using isotope geochemistry, chemical finger-printing, and modeling to differentiate coal ash contamination from other contaminant sources in the environment (e.g., see refs [42–44]). This is a very difficult task due to the extraordinary complexity of environmental chemistries and physical matrices. A unique fingerprint to trace the distribution of coal ash components around the globe is still lacking. More specific tracers are also needed for medical diagnoses, especially in the study of lung disease due to the severity of this problem as discussed in the Introduction of this article.

Therefore, Magnéli phases are presented here as a new tracer candidate for coal-burning power plant and likely coke production/smelting plant emissions, both fluvial and aeolian. Such a tracer could be used alone, or in combination with other source apportionment techniques mentioned above. Perhaps the strength of a Magnéli phase tracing method is in its surprising simplicity and ease of use. This stems from the confluence of a number of fortunate occurrences and processes that work together to provide such a method. $TiO_2$ mineral phases are essentially a ubiquitous accessory phase in all coals worldwide, generally varying from a tenth of a percent to several percent, by weight. Combustion of the coal, with included $TiO_2$ minerals which are never intentionally targeted for separation from the coal due to their relatively benign nature, and at temperatures over 1000 °C under locally reducing conditions, very quickly produces Magnéli phases. Although worthy of a separate study, the grain sizes of the Magnéli phases generated should be quite variable, although in this study, using TEM as the critical identification and tracking tool, we have selected for submicron grains. These Magnéli phases are then entrained with the many

other gaseous and condensed matter components of the plant emission stream. Some fraction of these phases will leave the plant with the gaseous stream (from virtually none to a significant proportion, depending on the plant design and efficacy of its emission-control systems). The remainder will end up in the collected ash components, to be later stored in impoundments or landfills, or recycled into various ash-containing products (concrete/cement, wallboard, road fill, etc.). In all of these possibilities, the incidentally produced Magnéli phases are entering some aspect of the environment, and are immediately or eventually (through secondary processes, e.g., wallboard disposal in a landfill, or incinerated) subject to transport via natural aeolian or fluvial process, and often a combination of the two.

The organismal and ecological environments that these Magnéli phases enter lack the occurrence of Magnéli phases, given their extraordinary rarity in natural systems. On the other hand, we estimate the production of Magnéli phases due to industrial coal-burning to be enormous over the last two centuries since the Industrial Revolution. Considering ballpark, but well-founded estimations of coal productions in modern times[1], accounting for the great majority of coal that has ever been used, one can say that on the order of a few hundred gigatons (Gt) of coal has been burned by humans. We can also estimate from existing observations and data that coal contains on the order of an average of 2% $TiO_2$, and that, on average, a coal will produce roughly 15% residual ash (both of these estimates are ours, from our personal estimations from the vast coal mineral and ash literature, and our own experience). While the conversion efficiency of $TiO_2$ minerals to Magnéli phases in the burning process is not known, we can say that, at least in the size range of up to several hundred nanometers (as observed in this study among 22 ashes collected from around the United States and China), most of the titania has been converted to Magnéli phases.

Considering all of this, we are still left with on the order of a Gt of Magnéli phase production via industrial coal-burning over the last few centuries to be spread via the atmosphere and hydro-sphere around the planet. The bottom line is that if one observes these highly unusual, easily recognizable phases in selective searching with a TEM, whether in or on soils, river/estuary water or sediments, city streets, stormwater ponds, human lung tissue, or wherever else, that portion of the environment has been very likely impacted by these phases and other solid state components in a similar size range from the sources that have been described herein. The reach of these small physical components (e.g., naturally occurring nano-sized inorganic solids) is global, as is well-known and documented[10].

Until now, and as described in detail above, titania suboxides were not known to be widely distributed in the environment, and to our knowledge have not been used in commercial products with wide distribution[20]. This explains why toxicity testing had not been initiated and underscores why an understanding of the particles' toxicity potential is now needed. In comparison, the nanoparticles of the polymorphs of $TiO_2$ are widely used in commercial products, from paints to makeup to sunscreens, among other uses, and as expected, the toxicity of these nano-particles have been widely studied. $TiO_2$ nanoparticle toxicity has been extensively examined in the context of freshwater and marine environments and in organisms belonging to several trophic levels[45–48]. It is generally thought that oxidative stress is the primary mechanism of acute toxicity to developing organisms due to the ability of forms of $TiO_2$ to produce reactive oxygen species (ROS) when irradiated[49–52]. ROS leads to cell death via protein, lipid, and DNA damage[45, 46, 48]. There is also evidence that sub-chronic exposure to adult zebrafish causes neurological effects associated with spatial recognition memory and

behavior[52]. Several studies implicate $TiO_2$ nanoparticle coating, shape, size, and aggregation properties as confounding variables to these nanoparticle's bioavailability and toxicity[45, 46, 50, 53]. This is especially relevant in studies using zebrafish models because of the protective chorion (an acellular enclosure) encompassing the developing fish between 0–48 hpf. Though its permeability has not been thoroughly characterized, the zebrafish chorion is a known barrier to the uptake of many contaminants, including metallic nanomaterials[29, 54–57]. Studies aiming to characterize the toxicity of metal nanoparticles commonly remove the chorion prior to exposure to ensure particle bioavailability during early stages of development when organisms are most sensitive to environmental stressors[29, 54, 56, 57]. In fact, in additional experiments with chorionated zebrafish embryos in this study (data not shown), we failed to observe significantly reduced survival after exposure to P25 and P25-900 °C nanoparticles at concentrations up to 1000 ppm; these results corroborate the references cited above, demonstrating the protective effect of the zebrafish chorion barrier to titanium oxide nanomaterial bioavailability. The chorion is also commonly removed in screening assays investigating the toxicity potential of pharmacological agents or other chemicals; for these purposes, the dechorionated embryonic zebrafish is a high-throughput and robust vertebrate model[54, 56, 57].

Therefore, in this first study of titania suboxide toxicity, we were primarily interested in reporting the inherent toxicity of the P25-900 °C (Magnéli phase) particles in the absence of the embryonic zebrafish protective chorion barrier (i.e., toxicity relevance to aquatic animal gills or terrestrial animal lungs). We observed significant reductions in embryo survival following acute exposure to these particles in the absence of SSR, but not following co-exposure to P25-900 °C and SSR. These results are in stark contrast to those of P25, in which the detrimental effects are primarily attributed to the material's photo-induced toxicity due to the generation of ROS as described above. In contrast, the reason for the toxicity of Magnéli phase titania suboxides is not known, this being the first time that a formal biotoxicity study has been conducted for these materials. Whatever role Magnéli phases play in and around biological systems is expected to be fundamentally different than $TiO_2$ phases. Unlike the wide band gap semiconducting, UV absorbing $TiO_2$ phases, Magnéli phases are narrow band gap semiconductors (see above), with electrical conducting properties similar to carbon/graphitic materials[20]. Additionally, these variably defective materials are expected to have interesting and important ionic conductivity and catalytic properties that still need to be thoroughly explored.

The immediate relevance of this work is aquatic organism exposure under conditions of limited solar radiation co-exposure, but there are also translational implications for human health. Of particular interest is the reactivity of Magnéli phases in human lung alveolar membranes considering their accessibility to this critical and vulnerable portion of our physiology, and the fact that their biological reactivity likely does not depend on solar radiation co-exposure.

Overall, Magnéli phases are a newly recognized incidental nanomaterial in the environment with very wide distribution, allowing it to be a new candidate for tracing the distribution of industrial coal-burning solid-state emissions worldwide. In addition, while we provide an initial assessment of Magnéli phase toxicity using an established exposure regimen and animal model, this situation also clearly invites further toxicity studies aimed at comprehensive characterization of Magnéli phase toxicity at sublethal concentrations and investigations of potential mechanisms of action. New toxicity pathways in micro- to macro-organisms, including humans, are likely to be found.

## Methods

**$Ti_xO_{2x-1}$ discovery in coal ash.** As explained in the main body of this article, we surmised the incidental occurrence of $Ti_xO_{2x-1}$ in coal ash when we observed this phase in river sediment impacted by the Duke Energy coal ash spill on 9 February 2014. Samples of the impacted downstream riverbed sediment, the sediment upstream from the spill site, as well as the coal ash at the power station, and samples of unburned coal at the plant, were collected. Samples were analyzed via $HNO_3$–$HClO_4$–HF digestion and using a Thermo Electron X-Series ICP-MS. Ti oxides in samples were firstly investigated using an environmental scanning electron microscope (FEI Quanta 600 FEG) equipped with an energy-dispersive X-ray spectrometer (EDS, QUANTAX 400, Bruker). The backscattered electron mode was used to provide visual information based on the contrast between phases of different atomic numbers (Z). Selected Ti oxide particles in samples were further investigated using a scanning transmission electron microscope operating at 200 kV and equipped with a silicon drift detector-based EDS system (JEM 2100 TEM/Scanning-TEM, JEOL Corporation). Electron diffraction patterns of the crystalline and semi-crystalline phases were recorded in selected area electron diffraction (SAED) mode.

**Synthesis and analysis of $Ti_xO_{2x-1}$.** Fabrication of the Magnéli phase $Ti_xO_{2x-1}$ was carried out in a tube furnace with an $N_2$ atmosphere, and the $TiO_2$ nanoparticles (P25, anatase $TiO_2$, or rutile $TiO_2$) were located downstream of pulverized coal. The heating and cooling rate was 5 °C min⁻¹, and isothermal at target temperature for 2 h. The $N_2$ flow rate was 0.28 $m^3$ min⁻¹, and the tube diameter was 8.9 cm. P25 $TiO_2$ nanoparticles were also annealed in the tube furnace under 0.14 $m^3$ min⁻¹ $H_2$ flow rate with 0.28 $m^3$ min⁻¹ $N_2$ as the carrier gas. The heating and cooling rate for annealing at the $H_2$ atmosphere was also 5 °C min⁻¹, and isothermal at the target temperature for 2 h.

All the samples, including coal ash and environmental samples, were dispersed in water using an ultrasonic bath and further placed onto a 300-mesh copper TEM grid with a lacey carbon support film (Electron Microscopy Sciences, PA). Morphology, aggregation, and the diffraction patterns of the Ti oxides were investigated using a JEM 2100 TEM/STEM (JEOL Corporation) operating at 200 kV and equipped with an energy dispersive X-ray (EDX) spectrometer. Electron diffraction patterns of the crystalline and semi-crystalline phases were recorded in SAED mode.

XRD analyses were performed at a scanning rate of 5° min⁻¹ using a PANalytical X'Pert X-ray diffractometer (Almelo, Netherlands). Thermostability of $Ti_xO_{2x-1}$ was characterized by TGA (TA Instruments Q50), which recorded the weight loss of the Magnéli P25-1100 °C sample when subjected to a 5 °C min⁻¹ temperature ramp and isothermal at 250 °C for 7 days under ambient atmosphere. Raman spectra were measured using a Raman spectrometer (Senterra, Bruker Corporation) with a laser excitation of 532 nm. UV–visible absorbance was achieved by a UV–Vis spectrophotometer (Hitachi U-4100).

The photocatalytic performances of the coal-annealed titanium oxides were carried out with 4 ml of 10 mg l⁻¹ methylene blue aqueous solution and 1.2 mg coal-annealed samples, and the irradiation was from one sun AM 1.5 (100 mW cm⁻²) illumination with a solar simulator (150 W Sol 2A™, Oriel). The concentration change of methylene blue was monitored by the change of the 664 nm absorption peak.

Electronic band structures of the Magnéli phase $Ti_xO_{2x-1}$ were calculated using the plane-wave projector-augmented-wave method[58] within the framework of DFT, as implemented in the Vienna Ab initio Simulation Package[59]. We used the generalized gradient approximation parameterized by Perdew et al.[60] as exchange-correlation functional. The DFT + U approach with $U = 5$ eV applied to $3d$ states of Ti was used to take into account the electronic correlations in these materials containing $Ti^{3+}$ (with partially occupied $d$ bands). We did not consider the spin polarization for the Ti and O atoms, which is feasible for low temperature conditions of Magnéli phase $Ti_xO_{2x-1}$. Structural optimizations were done with the kinetic energy cutoff of 520 eV with $k$-point meshes of spacing $2\pi \times 0.03$ Å⁻¹, ensuring the residual forces smaller than $1 \times 10^{-3}$ eV Å⁻¹. The band structures were calculated by using the hybrid Heyd-Scuseria-Ernzerhof functional[61] with 25% exact Fock exchange, which remedies the inherent problem of standard DFT calculations underestimating band gaps of materials.

**Toxicity study.** For the toxicity assays in this study, we used synthesized Magnéli phases (as described above) and Aeroxide P25 (approx. 80/20 anatase/rutile; obtained from Evonik Degussa GmbH, Düsseldorf, Germany), at a concentration of 100 ppm. In all, 1 g l⁻¹ stock suspensions were prepared by suspending nanoparticles in DI $H_2O$ and sonicating in a bath sonicator for 60 min immediately prior to dosing. Control embryos were exposed to an equal volume of DI $H_2O$.

Zebrafish embryos (*Danio rerio*) were collected immediately after natural spawning of Ekkwill adult zebrafish (Ekkwill Waterlife Resources, Ruskin, FL, USA) and kept at 28 °C in Danieau water[62]. At 24 hpf, embryos were screened for development stage, dechorionated in 1 mg ml⁻¹ pronase, and rinsed three times with fresh Danieau[62, 63]. Dechorionated embryos were then either exposed to nanoparticles and SSR or control conditions between 25–27 hpf and survival was assessed at 48 hpf. SSR co-exposure was administered using an Atlas SunTest CPS + solar simulator equipped with a standard solar filter (Atlas Material Testing Solutions, Chicago, IL). The solar standard filter achieves outdoor solar radiation as

specified by DIN 67501, an international standard for SSR[64]. Plates intended to receive SSR were wrapped in Saran Wrap to allow UV penetration while plates serving as no-solar-radiation controls were subsequently wrapped in aluminum foil. All SSR exposures were performed at 250 watts m$^{-2}$ lamp output at 28 °C for 2 h to achieve total irradiation of 1800 kJ m$^{-2}$. Embryos were subsequently raised at 28 °C on a 14:10 h light:dark cycle. The experiment was repeated six times with 16 embryos used per treatment per replicate point.

Statistical analyses were completed using GraphPad Prism Software (Version 7.0, La Jolla, CA). The effect of nanomatieral exposure and solar radiation on embryo survival was tested by 2-way analysis of variance and Tukey's post hoc test was used to conduct multiple pairwise comparisons. An alpha level of 0.05 was used for all analyses.

**Data availability**. The authors declare that all data directly supporting the findings of this study are available within the paper and the Supplementary Information file. Any additional data are available from the corresponding author (M.H.) upon request.

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

# ARTICLE

50. Faria, M., Navas, J., Soares, A. & Barata, C. Oxidative stress effects of titanium dioxide nanoparticle aggregates in zebrafish embryos. *Sci. Total Environ.* **470-471**, 379–389 (2014).

51. George, S. et al. Differential effect of solar light in increasing the toxicity of silver and titanium dioxide nanoparticles to a fish cell line and zebrafish embryos. *Environ. Sci. Technol.* **48**, 6374–6382 (2014).

52. Sheng, L. et al. Mechanism of TiO₂ nanoparticle-induced neurotoxicity in zebrafish (Danio rerio). *Environ. Toxicol.* **31**, 163–175 (2016).

53. Wyrwoll, A. et al. Size matters- the phototoxicity of TiO₂ nanomaterials. *Environ. Pollut.* **208**, 859–867 (2016).

54. Mandrell, D. et al. Automated zebrafish chorion removal and single embryo placement: Optimizing throughput of zebrafish developmental toxicity screens. *J. Lab. Autom* **17**, 66–74 (2012).

55. Shih, Y. et al. Adsorption characteristics of nano-TiO₂ onto zebrafish embryos and its impacts on egg hatching. *Chemosphere* **154**, 109–117 (2016).

56. Kim, K. & Tanguay, R. The role of chorion on toxicity of silver nanoparticles in the embryonic zebrafish assay. *Environ. Health Toxicol* **29**, e2014021 (2014).

57. Wehmas, L. et al. Comparative metal oxide nanoparticle toxicity using embryonic zebrafish. *Toxicol. Reports* **2**, 702–715 (2015).

58. Bloechl, P. E. Projector augmented-wave method. *Phys. Rev. B.* **50**, 17953 (1994).

59. Kresse, G. & Furthmueller, J. Efficient iterative schemes for ab initio total-energy calculations using a plane-wave basis set. *Phys. Rev. B.* **54**, 11169 (1996).

60. Perdew, J. P., Burke, K. & Ernzerhof, M. Generalized gradient approximation made simple. *Phys. Rev. Lett.* **77**, 3865 (1996).

61. Heyd, J., Scuseria, G. E. & Ernzerhof, M. Hybrid functionals based on a screened Coulomb potential. *J. Chem. Phys.* **118**, 8207–8215 (2013).

62. Nasevicius, A. & Ekker, S. C. Effective targeted gene 'knockdown' in zebrafish. *Nat. Genet.* **26**, 216–220 (2000).

63. Westerfield, M. *The Zebrafish Book. A Guide for The Laboratory Use of Zebrafish (Danio Rerio)*, 4th edn. (Univ. of Oregon Press, 2000).

64. E.V. Deutsches Institut für Normung (EV DIFN). *Experimental In Vivo Evaluation Of The Protection From Erythema Of External Sunscreen Products For The Human Skin. DIN 67501* (Deutsches Institut für Normung e.V., 2010).

## Acknowledgements

This study was supported by the National Natural Science Foundation of China (41522111; 41271473 and 41130525). Additional funding for this work was provided by the Center for the Environmental Implications of Nanotechnology (NSF Cooperative Agreement EF-0830093), the Virginia Tech Institute for Critical Technology and Applied Science (ICTAS), the Open Foundation of East China Normal University, the Virginia Tech National Center for Earth and Environmental Nanotechnology Infrastructure (NSF Cooperative Agreement 1542100), the National Institute of Environmental Health Science (NIEHS) (Training Grant #T32-ES021432), the STAR Fellowship Assistance Agreement no. FP-91780101-1 awarded by the U.S. Environmental Protection Agency (EPA), and the Recruitment Program of Global Youth Experts in China. B.C. and S.P. also acknowledge the financial support from the Office of Basic Energy Science, Department of Energy (DE-FG02-09ER46674). This paper has not been formally reviewed by EPA. The views expressed in this publication are solely those of the authors and EPA does not endorse any products or commercial services mentioned in the publication. We thank Jeffrey Parks for the ICP-MS analysis, Stephen McCartney in the Nanoscale Characterization and Fabrication Laboratory at Virginia Tech, Ben Coleman at Duke University (now at the University of Montana), Weinan Leng at Virginia Tech, Yiren Zhu at JSNU, and Qianqian Fu, Dengpeng Lan, Xin Zheng and Feiyun Tou at ECNU.

## Author contributions

Y.Y. and M.F.H. initiated and designed this study. Y.Y. performed TEM analyses of all the samples; B.C. performed the synthesis of Ti oxides, thermos-stability, and photo-reactivity of the materials advised by S.P. C.W. performed diffraction analyses using double tilt TEM holders. J.H. collected and selected coal ash samples and M.L. collected samples from China for TiOx investigation. M.S. worked out the crystallography of the Ti oxides observed in this study, and made the final conclusive identifications. Y.C. performed the annealing experiments using sludges and P25, advised by J.G. J.B. executed the toxicity studies, advised by R.D.G. Y.F. and L.Z. carried out the simulation of Ti oxides. The manuscript was written by M.F.H., Y.Y., B.C., J.B., M.S. and J.H. and edited by M.F.H.

## Additional information

**Competing interests:** The authors declare no competing financial interests.

