## [Peer Review File · Nature Communications]

Reviewers' Comments:

Reviewer #1 (Remarks to the Author)

Review of "Discovery and ramifications of incidental Magnéli phase generation and release from industrial coal burning" by Y. Yang et al.

This is a generally well-written manuscript characterizing nano-scale titania suboxides found for the first time in association with coal combustion products stored in the environment. TiO₂ polymorphs are a common minor constituent of coal and other sediments and the case presented by the authors for the existence of titania suboxides formed at boiler temperatures is convincing. But there are really two papers here. One paper should include the following 1) Identification of Magnéli phases in coal combustion products; 2) Detailed characterization of these Magnéli phases including experimental studies and any information in the literature pertaining to their conditions of formation; and 3) Discussion of their potential distribution in the environment, including use as a possible tracer for coal combustion. Together, these topics should be sufficient for the paper to be publishable. A separate stand-alone study is needed on the toxicology of these particles. The section "Initial toxicology study of Magnéli phases" (p. 14-15) is a start, but in my opinion, this brief add-on will not cut it with the medical community who would be the primary audience for the second paper. I recommend that the preliminary section on toxicology be deleted or you can summarize it in the discussion section as preliminary results. Other specific comments follow.

P3 L20 Not sure it is correct to say that more attention has been paid to ultrafine particles than PM_{2.5} itself. Could say that less attention has been paid to ultrafines but they are potentially more harmful.

P3 L24 "coal continued to surge forward" is anthropomorphic. Can say that world coal use continues to increase.

P4 L18 Give references here for occurrence of TiO₂ polymorphs in coal.

P5 L1 Specify what conditions are present in coal burning power plants at which Magnéli phases are formed.

P6 L1-3 Where Magnéli phases were synthesized, specify conditions of formation.

P8 and elsewhere, Can you distinguish formation of Magnéli phases from TiO₂ polymorphs vs. addition of tramp steel in the coal grinding process, rather than the coal itself?

P11 L6-7 Again, give conditions of Magnéli phase formation where found in other settings.

P16 L5 Compare conditions of Magnéli phase synthesis in experiments vs. those present in coal-burning power plants.

P18 L1-8 Consider potential use of Magnéli phases as a coal trace together with other proposed tracers, such as mercury isotopes.

P19 L1-23 Speculative. Can condense this section into a single discussion point.

Recommendation: Minor Revisions

Reviewer #2 (Remarks to the Author)

The paper describes the discovery of Ti suboxide Magnéli phases in river sediment impacted by coal fly ash, and subsequent studies that aimed to identify their source, mechanism of formation, physical and toxicological properties. The data are of very high quality, and the manuscript demonstrates a thoughtful and thorough approach toward a better understanding of the occurrence and ramifications of incidental Magnéli phase generation and release from industrial coal burning. I believe the manuscript is of potential interest to the journal readership, but I have a number of comments that could be considered before arriving at a final decision.

Editorial/style Comments

2nd paragraph, page 4: The authors refer to "...the story presented in this article...". It may be my personal bias, but I tend to think of a story as a form of literature that is unconstrained by facts. I would refrain from using this descriptor for technical subjects. In the same sentence, the authors refer to the Magnéli phases as "...a direct incidental nanomaterial consequence of coal...". I understand their meaning, but on the surface the terms direct and incidental appear to be a contradiction. 3rd paragraph, page 4: Awkward sentence: We originally made the discovery of incidental Magnéli phases and presumed its generation associated with coal combustion during an investigation of a coal ash spill into a riverine environment. I suggest the following: We originally made the discovery of incidental Magnéli phases during an investigation of a coal ash spill into a riverine environment and presumed its generation is associated with coal combustion. 2nd paragraph, page 8: Suggest the following changes to sentence: Three examples of these Magnéli phases are shown in Fig. 3, two of these samples are from the United States and one is from China. 4th from last line, page 11: Duplicate word: "The X-ray ray diffraction (XRD) pattern..." 2nd line, page 12: Correction: "...and the dominate dominant phase for coal-annealed..." Last line, page 22: Correction: "...stock suspensions were prepared by dissolving suspending NPs in DI H₂O..." Table S1, Heading for Column 4: Correction: Largest D d spacing

Scientific Comments Page 5, 1st Paragraph in Results: The authors report Ti concentrations of "...4.6-5.2 g/kg (in river sediments upstream from the spill site), and 4.7-6.1 g/kg (in river sediments downstream from the spill site)." and then they state that "Compared to the unburned coal, elevated Ti concentrations in coal ash samples suggest that coal ash is very likely a significant source for Ti in the downstream sediments." It's a minor issue, but the use of the word "significant" could be questioned here. The range of reported Ti concentrations downstream is higher than upstream, but is this significant? Possibly, but the strength of this assertion could be supported with statistical tests of significance on groups of samples collected upstream versus downstream. 4th and 5th from last line on page 18; 3rd and 4th from last line on page 19: A key point made in the paper is that the dominant source of these Magnéli phases to the environment is from coal-burning facilities. I do not dispute this, but I point out there may be some other sources that have not been fully explored in the text. For example, it is duly pointed out on page 18 that they may be "...recycled into various ash containing products (concrete/cement, wallboard, road fill, etc.)". Of these examples, it might be expected that Magnéli phases would be essentially immobilized in concrete or road fill, but at the end of their term of use, wallboard or other recycled materials are commonly disposed of by incineration which could lead to secondary dispersion. The author's state that "...the nanoparticles

of the polymorphs of TiO₂ are widely used in commercial products, from paints to makeup to sunscreens, among other uses,...” and this points to another potentially important source of Magnéli phases to the environment. When these products enter the waste stream, incineration is a very common waste-disposal method worldwide and the fate of nano particulates is of interest (Buha et al. 2014, Environ. Sci. Technol.48; 4765–4773). If this potential source turns out to be non-trivial, what does it mean for the proposal that Magnéli phases could represent an effective tracer for coal-burning emissions...? The authors identified Magnéli phases in road dust from China, a nation known for extensive coal burning, so the inference that Magnéli phases in the road dust are derived from coal ash seems reasonable. However, a quick search for data on metal concentrations in aerosols suggests that Ti concentrations are negatively correlated with coal burning (Zhang et al. 2015, Atmospheric Pollution Research, 6; 635-643). It would be good to see some discussion that draws from the literature on aerosols from coal-burning plants and waste incineration. Tom Al Department of Earth & Environmental Sciences University of Ottawa November 21 2016

Reviewer #1 (anonymous):

This is a generally well-written manuscript characterizing nano-scale titania suboxides found for the first time in association with coal combustion products stored in the environment. TiO₂ polymorphs are a common minor constituent of coal and other sediments and the case presented by the authors for the existence of titania suboxides formed at boiler temperatures is convincing. But there are really two papers here. One paper should include the following 1) Identification of Magnéli phases in coal combustion products; 2) Detailed characterization of these Magnéli phases including experimental studies and any information in the literature pertaining to their conditions of formation; and 3) Discussion of their potential distribution in the environment, including use as a possible tracer for coal combustion. Together, these topics should be sufficient for the paper to be publishable. A separate stand-alone study is needed on the toxicology of these particles. The section “Initial toxicology study of Magnéli phases” (p. 14-15) is a start, but in my opinion, this brief add-on will not cut it with the medical community who would be the primary audience for the second paper. I recommend that the preliminary section on toxicology be deleted or you can summarize it in the discussion section as preliminary results. Other specific comments follow.

First, we really appreciate this anonymous reviewer’s careful study of our manuscript, and for challenging us (in the paragraph above) on the appropriateness of the last major part of the Discussion section of this paper, entitled “TiO₂ and Magnéli phase toxicity”.

We feel strongly that it is essential to keep this section on zebrafish toxicity in this manuscript. We do, however, take the blame for not being clearer as to why this section is so vitally important (we have made these modifications in the revised paper, as noted specifically below).

These zebrafish experiments are definitely not a “brief add-on” to our study, as the

reviewer stated. It took the better part of a year for very close colleagues of ours, co-authors Brandt and Di Giulio at Duke University, to obtain these data. These experiments take superb expertise and experience, and the work is long and tedious in order to make absolutely sure that the data is highly reliable. Prof. Di Giulio is an exceptionally accomplished and noted ecotoxicologist who we have worked with since 2007 through the NSF-funded CEINT project (Center for Environmental Implications of NanoTechnology). His senior PhD student, Jessica Brandt, is an expert zebrafish toxicologist, and well as a superb ecologist.

It is true that these experiments as reported in our paper do not begin to comprise a full characterization of the toxicity of Magnéli phase titanium oxides. Such a characterization, just with the zebrafish model, will require numerous studies over several years of time. In fact, this is ongoing with commercial TiO₂ nanomaterials by several groups over a number of years. But when designing this project, we predicted that the comprehensive characterization of the Magnéli phase particles and the discussion of their wide (seemingly ubiquitous?) distribution would stimulate questions regarding their toxicity. Therefore, the primary motivation for including the toxicity studies in this manuscript was to provide an initial understanding of their effects on embryonic fish survival using the well-established zebrafish (*Danio rerio*) model and to compare these effects to those elicited by well-studied TiO₂ particles.

The data described here provide a significant starting point for future investigations. They reveal a major difference in the biological activity of these two forms of titanium oxides with the Magnéli phase having greater activity in the dark versus TiO₂ having greater photoactivated activity, a well-known characteristic of TiO₂. This has important ramifications that will influence future investigations in the “medical community”, as correctly and aptly pointed out by the reviewer. Specifically, that is to say that the very likely most important route of human exposures to Magnéli phase materials is inhalation, which would result in cellular exposures in the dark. In aquatic systems, sediments are likely to be the major repository for these phases, again where solar radiation is very limited. As opposed to TiO₂ for which studies in combination with solar radiation have dominated, studies with the Magnéli phase should include and perhaps be dominated by studies in the absence of solar radiation. Moreover, our results motivate fundamental studies to determine the mechanisms underlying Magnéli phase toxicity. All of these studies will take years, with these studies stemming from our labs and those of many other groups. But at least this present paper is setting a solid, dependable starting point, and a path, that was not preciously available. We are confident that this is a key contribution.

As we noted above, the reviewers critical comment was so important to us because we have strengthened wording around this last section of the Discussion, to emphasize the importance of this first look at Magnéli phase toxicity.

Here are the changes that we have made in the manuscript to set-up and emphasize the importance of this initial toxicity testing:

- On lines 129-130, we specifically pointed out the “well-established zebrafish (*Danio rerio*) embryo model” in the last line of the Introduction to the paper, to help emphasize at the start the grounding of this first toxicity testing exercise for Magnéli phases.
- On lines 515-516, at the very beginning of the Discussion section on toxicity, we have added what “underscores why an understanding of the particles’ toxicity potential is now needed.”
- In the last 11 lines of this Discussion section on toxicity (lines 538-549), we have moved a

sentence and added two additional sentences that makes it clear that our findings with the zebrafish model has “translational implications for human health”, that we are providing “an initial assessment of their toxicity”, and that “this situation also clearly invites further toxicity studies aimed at comprehensive characterization of Magnéli phase toxicity at sublethal concentrations and investigations of potential mechanisms of action”. This is added very much in the spirit of the reviewer’s consideration of this section of the paper, for which we are most grateful.

Additional comments (and responses) from Reviewer #1:

P3 L20 Not sure it is correct to say that more attention has been paid to ultrafine particles than PM_{2.5} itself. Could say that less attention has been paid to ultrafines but they are potentially more harmful.

We thank the reviewer for noticing this. A great deal of attention has been given to the ultrafines, for the reason given in the next sentence and reference 7. Many have published on ultrafines in other airborne materials. Nevertheless, we have deleted “more attention” and simply say that ultrafine PM has been studied for the reason already given in the text. This change can be found on lines 66-67.

P3 L24 “coal continued to surge forward” is anthropomorphic. Can say that world coal use continues to increase.

Thank you for the suggestion and we have modified the sentence on line 70 to avoid the anthropomorphism.

P4 L18 Give references here for occurrence of TiO₂ polymorphs in coal.

Thank you for the suggestion and we have added several references (6) here as suggested. This change can be found on lines 104-105.

P5 L1 Specify what conditions are present in coal burning power plants at which Magnéli phases are formed.

This comment pertains to lines 124-130, the last paragraph of the Introduction. We feel that we should not specify the conditions under which Magnéli phases are produced in this sentence and at this stage of the paper. This paragraph very quickly touches on all critical aspects of the paper that will follow, just to tell the reader what is coming. Giving specifics on the conditions of Magnéli phase production, as the reviewer is requesting here, and not all the other accompanying parts of the paper, would be unbalanced and awkward. The sentence does specify that Magnéli phases are “generated during coal combustion”, so at least the reader knows that we are talking about high temperature environments.

P6 L1-3 Where Magnéli phases were synthesized, specify conditions of formation.

We have done exactly this with a new sentence (and three new supporting references) in lines 171-173 in the revised manuscript. Thanks for the suggestion. This adds nicely to our understanding and description of Magnéli phases in the manuscript.

P8 and elsewhere, Can you distinguish formation of Magnéli phases from TiO₂ polymorphs vs. addition of tramp steel in the coal grinding process, rather than the coal itself?

Tramp steel, used to grind the coal into a coarse powder that is then blown into the coal power plant firebox for burning, does not contain (of course) TiO₂ polymorphs. This should be completely self-evident to all readers. The only source of the TiO₂ polymorphs is the coal itself, as explained already prominently in the Results and also the Discussion sections of the paper (lines 258-267, and 391-402, and elsewhere). Therefore, we have not made any changes due to this comment.

P11 L6-7 Again, give conditions of Magnéli phase formation where found in other settings.

As suggested, the formation conditions, including temperatures and pressures, have been added. For the very rare geologic occurrence of Magnéli phases found in a “mudstone xenoliths”, the explicit conditions of formation have been given in line 262 (in the Results), and in a bit more detail, in lines 447 and 455 (in the Discussion).

P16 L5 Compare conditions of Magnéli phase synthesis in experiments vs. those present in coal-burning power plants.

Thank you for the suggestion. We have added the exact conditions of synthesis in both our experiments and the firebox of a coal-burning power plant in lines 383 to 386.

P18 L1-8 Consider potential use of Magnéli phases as a coal trace together with other proposed tracers, such as mercury isotopes.

We appreciate the reviewer’s suggestion. An addition reference, this an excellent one using mercury stable isotope abundances, has been added to the two references already given concerning source apportionment and tracing techniques (line 460). An additional sentence has been inserted (lines 467-468) emphasizing that a Magnéli phase tracing method could be used in combination with the isotope and modeling methods presented in the three references listed.

P19 L1-23 Speculative. Can condense this section into a single discussion point.

We strongly feel that this critical part of our Discussion, which starts on line 490, is not speculation, but valuable insight (through a deep knowledge in these matters from some of our authors) which will be of particular interest to environmental systems scientists, and well as policy and regulation experts. The lines that the reviewer points out have been carefully reasoned and vetted. Another way to put this is that this full section is necessary to satisfy readers who will want a ballpark estimate of Magnéli phase generation due to coal burning (and coke production) to put the entire process described in this paper into an overall context of the amount of this material added to the environmental Earth system, and how and why it can be used as a tracer for environmental impact. This general context, then, leads perfectly into the last two sections of the Discussion, and ultimately the most important implication of this paper, the biotoxicity of Magnéli phases. Therefore, we appeal to the editor not to require condensation of the section noted by the reviewer.

Reviewer #2 (signed by Prof. Tom Al):

It is a pleasure to have our paper reviewed by Prof. Tom Al of the Univ. of Ottawa. He is well-known internationally and an outstanding environmental scientist in the fields of contaminant hydrology, mineralogy, and geochemistry, very much appropriate for reviewing this paper. Thank you.

The paper describes the discovery of Ti suboxide Magnéli phases in river sediment impacted by coal fly ash, and subsequent studies that aimed to identify their source, mechanism of formation, physical and toxicological properties. The data are of very high quality, and the manuscript demonstrates a thoughtful and thorough approach toward a better understanding of the occurrence and ramifications of incidental Magnéli phase generation and release from industrial coal burning. I believe the manuscript is of potential interest to the journal readership, but I have a number of comments that could be considered before arriving at a final decision.

Editorial/style Comments

2nd paragraph, page 4:

The authors refer to "...the story presented in this article...". It may be my personal bias, but I tend to think of a story as a form of literature that is unconstrained by facts. I would refrain from using this descriptor for technical subjects.

We thank Prof. Al for this comment. He is, of course, correct, and we have changed the word "story" to "study". This is on line 99 of the current marked-up manuscript.

In the same sentence, the authors refer to the Magnéli phases as "...a direct incidental nanomaterial consequence of coal...". I understand their meaning, but on the surface the terms direct and incidental appear to be a contradiction.

Thanks for pointing this out. We have defined "direct" and "indirect" incidental materials elsewhere (in other publications), and the complexity of the definitions are not needed here. Therefore, we have just removed the word "direct".

3rd paragraph, page 4:

Awkward sentence: We originally made the discovery of incidental Magnéli phases and presumed its generation associated with coal combustion during an investigation of a coal ash spill into a riverine environment. I suggest the following: We originally made the discovery of incidental Magnéli phases during an investigation of a coal ash spill into a riverine environment and presumed its generation is associated with coal combustion.

We agree, and have written two shorter sentences to more clearly make this point. The new sentences are on lines 108-110.

2nd paragraph, page 8:

Suggest the following changes to sentence: Three examples of these Magnéli phases are shown in Fig. 3, two of these samples are from the United States and one is from China.

The change has been made as suggested, and additional punctuation has been added to the sentence to organize these thoughts and allow for easier reading. This sentence is on lines 218 to 220.

4th from last line, page 11:

Duplicate word: “The X-ray ray diffraction (XRD) pattern...”

2nd line, page 12:

Correction: “...and the dominate dominant phase for coal-annealed...”

Last line, page 22:

Correction: “...stock suspensions were prepared by dissolving suspending NPs in DI H₂O...”

Table S1, Heading for Column 4:

Correction: Largest Δd spacing

All of these changes have been made.

Scientific Comments

Page 5, 1st Paragraph in Results:

The authors report Ti concentrations of “...4.6-5.2 g/kg (in river sediments upstream from the spill site), and 4.7-6.1 g/kg (in river sediments downstream from the spill site).” and then they state that “Compared to the unburned coal, elevated Ti concentrations in coal ash samples suggest that coal ash is very likely a significant source for Ti in the downstream sediments.”

It's a minor issue, but the use of the word “significant” could be questioned here. The range of reported Ti concentrations downstream is higher than upstream, but is this significant? Possibly, but the strength of this assertion could be supported with statistical tests of significance on groups of samples collected upstream versus downstream.

This is a good point, and the comment made us realize that this sentence needed significant rewording. We have worded it much more carefully and completely to be accurately reflective of the data. This edited sentence is found on lines 140 to 143, and now reads: “Compared to Ti concentrations of unburned coal vs. coal ash, and slightly elevated Ti concentrations in the downstream sediment samples vs. upstream, the data suggest that coal ash is a source for elevated Ti in the downstream sediments.”

4th and 5th from last line on page 18; 3rd and 4th from last line on page 19:

A key point made in the paper is that the dominant source of these Magnéli phases to the environment is from coal-burning facilities. I do not dispute this, but I point out there may be some other sources that have not been fully explored in the text. For example, it is duly pointed out on page 18 that they may be “...recycled into various ash containing products (concrete/cement, wallboard, road fill, etc.)”. Of these examples, it might be expected that Magnéli phases would be essentially immobilized in concrete or road fill, but at the end of their term of use, wallboard or other recycled materials are commonly disposed of by

incineration which could lead to secondary dispersion.

This worthy point from Prof. Al was indirectly referred to in the original wording of the manuscript. We have now specifically mentioned and explained “secondary processes” parenthetically in that sentence in the revised version, this on lines 486-488.

The author’s state that “...the nanoparticles of the polymorphs of TiO₂ are widely used in commercial products, from paints to makeup to sunscreens, among other uses,...” and this points to another potentially important source of Magnéli phases to the environment. When these products enter the wastestream, incineration is a very common waste-disposal method worldwide and the fate of nano particulates is of interest (Buha et al. 2014, Environ. Sci. Technol.48; 4765–4773). If this potential source turns out to be non-trivial, what does it mean for the proposal that Magnéli phases could represent an effective tracer for coal-burning emissions...?

Due to this comment, we have taken the liberty to add one last “mini-section” to the end of the Discussion (13 lines of additional text). The section’s subtitle is: “A note on municipal solid waste incineration; another pathway for Magnéli phase production”. We actually performed additional experiments to be able to much more precisely address the question posed by Prof. Al above. We have added the reference about waste incineration and nanomaterials that Prof. Al pointed out, as well as two other pertinent and excellent publications on this subject. The bottom line is that the projected production rate of Magnéli phases due to the incineration of municipal waste is estimated to be only a very small fraction of the production from coal-burning power plants.

The authors identified Magnéli phases in road dust from China, a nation known for extensive coal burning, so the inference that Magnéli phases in the road dust are derived from coal ash seems reasonable. However, a quick search for data on metal concentrations in aerosols suggests that Ti concentrations are negatively correlated with coal burning (Zhang et al. 2015, Atmospheric Pollution Research, 6; 635-643). It would be good to see some discussion that draws from the literature on aerosols from coal-burning plants and waste incineration.

Due to the comment before this last one, we have inserted into the manuscript a rough estimate of Magnéli phase generation due to waste incineration, and also estimated that it does not add significantly to that produced by coal-burning power plants. In addition, however, we did have a look at the Zhang et al. paper that Prof. Al brought to our attention in this last comment. This paper shows that Ti concentrations of airborne particulate matter in Guiyang, China seems negatively correlated with coal burning, at least in the 16 days during 2011 and 2012 when this data was collected. When the single power plant in the city is burning the most coal (winter), the Ti-concentration in aerosols for four consecutive days in the winter when the actual aerosol collection was conducted was lower than the four days of measurement in each of the other three seasons. However, our paper is not about Ti concentrations in aerosols, but the discovery of Magnéli phase generation and its potential toxicity. Ti is tied up in many ways in aerosols, and in the Zhang et al. study, we have no idea what that Ti distribution is among the various carriers of Ti. In order to comment on that paper, we would need to know, at least, the Magnéli phase concentration in the aerosols. And these Magnéli phases would not only be coming from the Guiyang coal burning power

plant, but plants throughout other areas of China, all burning various coals with various TiO_2 contents, and therefore various Magnéli phase contents. Also, weather patterns always play a big role in aerosol collection, especially when, as in the Zhang et al. paper, they used only a single collection point in the entire study (which we feel is an important flaw in this study).

As a result of our commentary above, we have chosen not to comment on this paper, or others like it, in this present manuscript. It would be unproductive without much more collected information. Such studies will be very important in the future, but for now, this is all well beyond the scope of our study.

Reviewers' Comments:

Reviewer #1:

Remarks to the Author:

With revisions, the authors have addressed the reviewer suggestions, but in doing so, have retained the original organization in which toxicity is considered as one of many aspects. I think the study can and should be published as is. But I still question whether the medical community will be willing to traverse all these other aspects to get to the toxicity study. I think the research team would be well served to follow with a separate study devoted entirely to the toxicity and potential human health impact of these materials.

Reviewer #2:

Remarks to the Author:

I believe the authors have carefully considered the review comments and I am satisfied with their responses. I commend them for their high-quality work.

Reviewer #3:

Remarks to the Author:

The previous reviewer commented on the validity of the section "Initial toxicology study of Magneli phases" suggesting that the zebrafish toxicity experiments were not sufficient to include. While the reported studies are limited, I support the authors' response that these toxicity experiments are a key contribution and provide an initial understanding of the effects of Magneli phases to a well-established zebrafish model. These initial studies are likely to encourage additional research of TiO₂ Magneli phases which are critical to our understanding of the potential toxicological effects of coal burning.

The toxicology component of this manuscript included two sets of experiments. The first monitored mortality up to 144 hpf of zebrafish embryos exposed to P25 Magneli phase nanoparticles and P25-900°C nanoparticles, with and without solar radiation. For each Magneli phase treatment, six concentrations were tested ranging from 0-1000 ppm. This experiment was replicated 10 times with 4 embryos per treatment per replicate. No mortality was observed in any of the treatments except 500ppm P25-900°C + SSR.

The second experiment measured mortality at 48 hpf of dechorinated embryos exposed to 100 ppm of P25 and P25-900°C, with and without SSR. Mortality was only significantly affected with solar radiation in P25 and without solar radiation in P25-900°C; no other reductions in mortality were observed. This experiment was repeated 6 times with 16 embryos per treatment per replicate.

Experiment 1 (exposed at 4 hpf; dose-response; terminated at 6 dpf):

1. Although repeated 10 times, the first experiment used 4 replicates per treatment with only 4 fish per treatment. This means that each fish was 25% of the sample for each replicate. This is highly unusual for toxicity testing. Guidelines for early life stage toxicity testing with fish (e.g., OECD 236) recommend using a minimum of 20 embryos per concentration, such that a 25% mortality rate would require the mortality of 5 individuals, not just 1. This, in concert with lack of a clear dose-response (i.e. no response at the highest concentration, not enough data to support a hormetic effect), makes me question the robustness of these data. There is no need to repeat a test which sufficient numbers of test organisms per replicate.

2. There is no presentation in the text or the figure for 'Experiment 1' what were the results for the control embryos. Even if the purpose is to contrast toxicity from exposures to nanoparticles, reporting control survival is necessary. In the OECD protocol, a minimum survival of 90% is

required for a test to be valid.

3. Please provide the test statistics for all reported significant differences (i.e. experiment 1: 500 ppm + SSR, Experiment 2: P25+SSR and P25-900°C).

Experiment 2:

1. The authors do not provide sufficient explanation for the second experiment's results. They indicate that their results contrast the effects of TiO₂ nanoparticles with solar radiation but do not provide any further discussion.

2. No explanation for why 100ppm concentration was selected.

3. 530-533: The authors explain that dechorinated embryos were used in this study because previous research has showed the protective effects of the chorion. More explanation is needed to provide justification for dechorination. Examples of other studies using dechorinated embryos would help, e.g. in the context of extrapolating to aquatic and terrestrial animals with no protective chorion.

Overall comments:

From a toxicology perspective, I think the results of Experiment 2 are robust and should be included as an initial exploration of the toxicity of the substances under discussion. However, I think that the data from Experiment 1 is not robust enough to include in the paper, as explained above.

The authors discuss the implications of exposure to Magneli phase particles to both aquatic and terrestrial animals (e.g., of course humans). Testing dechorinated embryos as they did in Experiment 2 is relevant toxicologically to aquatic animals with gills as well as terrestrial animals with lungs. Therefore, describing this test alone and discussing the results and implications is justifiable.

Testing embryos with intact chorions is however still important because of potential impacts to developing aquatic animals with chorions. Therefore, I would urge the authors to repeat their Experiment 1. In the authors' response to the first set of reviews, they indicated that these tests required too much time to repeat. If a zebrafish exposure is conducted with sufficient individuals per replicate, it need not be repeated and would take only one week to conduct Experiment 1 anew. If the particles must be generated anew and this is what takes too much time, then I recommend publishing only the results from Experiment 2. Edits to the text would obviously be required if Experiment 1 were to be excluded.

Reviewer 1:

Comment: With revisions, the authors have addressed the reviewer suggestions, but in doing so, have retained the original organization in which toxicity is considered as one of many aspects. I think the study can and should be published as is. But I still question whether the medical community will be willing traverse all these other aspects to get to the toxicity study. I think the research team would be well served to follow with a separate study devoted entirely to the toxicity and potential human health impact of these materials.

We appreciate Reviewer 1's comment that "the study can and should be published as is". However, he/she still questions whether the medical community will notice the toxicity implications because it is still just one of many other aspects of this paper. Based on this feedback, we voluntarily revised the framing of the experimental relevance in the paper to more clearly explain how the toxicity results relate to aquatic and non-aquatic exposure scenarios (including to human health). Considering Reviewer 3's perspective as well, these two reviewers agree with us that the values of the biotoxicity studies are key to the overall scientific thrust of our work.

We also clearly agree, and make it clear in the paper, that further research on the toxicity of these materials and greater exploration of human health risks are warranted and we will pursue funding for such work in the future. With our discoveries as presented in this paper, we know that many other groups will do the same. We have been in this business for a long time, and we know that our discovery will become a very active subfield of nanoparticulate human toxicology in the future.

We thank Reviewer 2 for his/her outstanding interest in our work. His/her comments have allowed for us to add additional clarity as to why future studies in the human biotoxicity of Magnéli phase generation from industrial coal burning are so important.

Reviewer 2:

Comment: I believe the authors have carefully considered the review comments and I am satisfied with their responses. I commend them for their high-quality work.

Again, we thank Reviewer 2 (Prof. Tom Al) for his input on the original version. The manuscript is improved as a result of his thoughtful feedback.

Reviewer 3:

Experiment 1 (exposed a 4hpf; dose-response; terminated at 6 dpf):

1. Although repeated 10 times, the first experiment used 4 replicates per treatment with only 4 fish per treatment. This means that each fish was 25% of the sample for each replicate. This is highly unusual for toxicity testing. Guidelines for early life stage toxicity testing with fish (e.g., OECD 236) recommend using a minimum of 20 embryos per concentration, such that a 25% mortality rate would require the mortality of 5 individuals, not just 1. This, in concert with lack of a clear dose-response (i.e. no response at the highest concentration, not enough data to support a hormetic effect), makes me question the robustness of these data.

2. There is no need to repeat a test which sufficient numbers of test organisms per replicate. There is no presentation in the text or the figure for 'Experiment 1' what were the results for the control embryos. Even if the purpose is to contrast toxicity from exposures to nanoparticles, reporting control survival is necessary. In the OECD protocol, a minimum survival of 90% is required for a test to be valid.

Our response to Reviewer 3's comments on Experiment 1, both points 1 and 2 above, can be found in our response to the Reviewer's last comments below, entitled "**Overall comments**".

3. Please provide the test statistics for all reported significant differences (i.e. experiment 1: 500 ppm + SSR, Experiment 2: P25+SSR and P25-900°C).

The test statistics for all reported significant differences are now included in the Results section of the paper in lines 2 to 7 on page 15. These results now read:

Dechorionated embryos co-exposed to P25-900°C + SSR were not different from controls, but those exposed to P25 + SSR exhibited very high mortality (adjusted P value <0.0001). In the absence of SSR, we observed a very different result. There, dechorionated embryos exposed to 100 ppm P25-900°C showed significant mortality (adjusted P value = 0.0004), but those exposed to P25 showed slightly reduced, but not significantly different survival (adjusted P value = 0.8626).

Experiment 2:

1. The authors do not provide sufficient explanation for the second experiment's results. They indicate that their results contrast the effects of TiO₂ nanoparticles with solar radiation but do not provide any further discussion.

The authors appreciate this request and have modified the discussion of the experiment's results that considers differences in the physico-chemical properties of the two nanomaterials as explanatory variables. This modified text (beginning on line 20, page 20, and ending on line 3, page 21) is as follows:

Therefore, in this first study of titania suboxide toxicity, we were primarily interested in reporting the inherent toxicity of the P25-900°C (Magnéli phase) particles in the absence of the embryonic zebrafish protective chorion barrier (i.e. toxicity relevance to aquatic animal gills or terrestrial animal lungs). We observed significant reductions in embryo survival following acute exposure to these particles in the absence of SSR, but not following co-exposure to P25-900°C and SSR. These results are in stark contrast to those of P25 (TiO₂ NP), in which the detrimental effects are primarily attributed to the

material's photo-induced toxicity due to the generation of ROS as described above. In contrast, the reason for the toxicity of Magnéli phase titania suboxides is not known, this being the first time that a formal biotoxicity study has been conducted for these materials. Whatever role Magnéli phases play in and around biological systems is expected to be fundamentally different than TiO₂ phases. Unlike the wide band gap semiconducting, UV absorbing TiO₂ phases, Magnéli phases are narrow band gap semiconductors (see above), with electrical conducting properties similar to carbon/graphitic materials²⁰. Additionally, these variably defective materials are expected to have interesting and important ionic conductivity and catalytic properties that still need to be explored.

2. No explanation for why 100ppm concentration was selected.

We have revised the results section for this experiment and explained the selection of 100 ppm concentration on lines 15 to 20 (on page 15) as follows:

Fig. 6 shows dechorinated zebrafish embryo survival at 48 hours post fertilization (hpf) in replicates exposed to P25 nanoparticles with and without concurrent simulated solar radiation (SSR), at titania concentrations of 100 ppm, as well as control groups with no titania exposure, all this modeled after Bar-Ilan et al.²⁹ for the sake of comparison. The only difference between the Bar-Ilan et al. study and the present one is that exposure to P25-900°C nanoparticles was also carried out.

3. 530-533: The authors explain that dechorinated embryos were used in this study because previous research has showed the protective effects of the chorion. More explanation is needed to provide justification for dechorination. Examples of other studies using dechorinated embryos would help, e.g. in the context of extrapolating to aquatic and terrestrial animals with no protective chorion.

Due to this comment, we have expanded upon our rationale for using dechorinated embryos in this experiment in the discussion section of the manuscript, now at lines 9 to 19, page 20, as follows:

Studies aiming to characterize the toxicity of metal nanoparticles commonly remove the chorion prior to exposure to ensure particle bioavailability during early stages of development when organisms are most sensitive to environmental stressors^{29,51,53,54}. In fact, in additional experiments with chorionated zebrafish embryos in this study (data not shown), we failed to observe significantly reduced survival after exposure to P25 and P25-900°C nanoparticles at concentrations up to 1000 ppm; these results corroborate the references cited above, demonstrating the protective effect of the zebrafish chorion barrier to titanium oxide nanomaterial bioavailability. The chorion is also commonly removed in screening assays investigating the toxicity potential of pharmacological agents or other chemicals; for these purposes, the dechorinated embryonic zebrafish is a high-throughput and robust vertebrate model^{51,53,54}.

Overall comments:

From a toxicology perspective, I think the results of Experiment 2 are robust and should be included as an initial exploration of the toxicity of the substances under discussion. However, I think that the data from Experiment 1 is not robust enough to include in the paper, as explained above.

The authors discuss the implications of exposure to Magnéli phase particles to both aquatic and terrestrial animals (e.g., of course humans). Testing dechorinated embryos as they did in Experiment 2 is relevant toxicologically to aquatic animals with gills as well as terrestrial animals

with lungs. Therefore, describing this test alone and discussing the results and implications is justifiable.

Testing embryos with intact chorions is however still important because of potential impacts to developing aquatic animals with chorions. Therefore, I would urge the authors to repeat their Experiment 1. In the authors' response to the first set of reviews, they indicated that these tests required too much time to repeat. If a zebrafish exposure is conducted with sufficient individuals per replicate, it need not be repeated and would take only one week to conduct Experiment 1 anew. If the particles must be generated anew and this is what takes too much time, then I recommend publishing only the results from Experiment 2. Edits to the text would obviously be required if Experiment 1 were to be excluded.

We greatly appreciate Reviewer 3's careful attention and thoughtful feedback on the toxicity experiments in this manuscript. All comments and suggestions regarding Experiment 2 have been considered and addressed in the revised version and each critique is individually responded to above.

After careful consideration of the feedback for Experiment 1, we have decided to remove this experiment from the manuscript. The reviewer's primary concern with this experiment was the small number of fish (4 embryos) constituting an $n = 1$. Instead, the reviewer suggests repeating the experiment a single time with a minimum of 20 embryos per treatment. This would present a logistical challenge as well as a substantial time commitment; we only used 4 embryos per treatment group in this experiment because of physical space limitations in the SunTest instrument we use to administer solar simulated radiation. This instrument fits four well-plates per dosing. For these experiments, we simultaneously ran four 48-well plates. Each plate set-up was identical, with two plates receiving SSR exposure and two plates wrapped in foil so as not to receive SSR but be subject to the same temperature variations as the SSR exposed treatment groups. While we could have increased the number of embryos per treatment group by using a 96-well plate, the time needed to plate an additional 192 embryos (48 embryos x 4 plates) would have delayed the exposure window. Our decision to use 48-well plates was, therefore, one intended to optimize the timing of SSR co-exposure between 4-6 hpf.

In the manuscript, we present the statistics per experimental plate in order to reflect that we used different groups of breeding embryos per plate in order to capture response differences that may be due to genetic variability. While this is not entirely necessary, it is common practice in our labs so that we can test for embryo cohort effects. In considering Reviewer 3's feedback, we re-ran the statistics such that all embryos from each treatment group (spread among two plates) were combined, providing 8 embryos per treatment group for five experimental repeats. Even still, we failed to observe a dose-response.

Repeating this experiment with at least 20 embryos for more than 20 treatment groups (2 nanomaterials at 5 concentrations, plus control groups, with and without SSR co-exposure), would take a considerable amount of time due to the space limitations of the SunTest instrument.

Taking this into consideration, along with the lack of dose-response using fewer embryos, and the emphasis on more interesting and certainly more relevant results (for this particular publication) in Experiment 2, we decided to drop Experiment 1 from the manuscript. Accordingly, we have made revisions to the text where appropriate. In the revised version, we simply acknowledge that an additional study of chorionated embryos was completed, and that it agreed with other published studies using chorionated embryos (lines 12-16, page 20).

Reviewers' Comments:

Reviewer #3:

Remarks to the Author:

The authors have addressed my concerns about Experiment 1. They describe limitations preventing them from repeating 'Experiment 1', and instead agreed to include only the more robust toxicology results of 'Experiment 2'. The authors have made appropriate changes to the methods, results, and discussion to reflect the smaller, but more robust data set for the toxicology testing. In my opinion, the manuscript now provides a defensible initial assessment of the toxicity testing of the Magneli phase particles, laying the groundwork and explaining the need for additional testing.